# Analysis of the Influence of the Vortex Shedder Shape on the Metrological Properties of the Vortex Flow Meter

**DOI:** 10.3390/s21144697

**Published:** 2021-07-09

**Authors:** Mariusz R. Rzasa, Beata Czapla-Nielacna

**Affiliations:** 1Faculty of Electrical Engineering, Automatic Control and Informatics, Opole University of Technology, Proszkowska 76 Street, 45-758 Opole, Poland; 2Faculty of Mechanical Engineering, Opole University of Technology, Mikolajczyka 5 Street, 45-271 Opole, Poland; beattacz@wp.pl

**Keywords:** vortex flow meter, flow visualization, flow measurement, flow visualization

## Abstract

Vortex flow meters are used to measure the flow of gases and liquids. The flow meters of this type measure the frequency of vortices that arise behind an obstacle set in the path of the flowing fluid. The frequency is a function of the speed of the flowing fluid. This obstacle is called the vortex shedder bar. The advantage of this solution is that the frequency of vortices does not viscose on the rheological properties of the fluid, such as viscosity or density. As a result, the indications of the vortex flowmeter do not depend on the temperature and type of fluid. The work includes numerical and experimental studies of the effect of changing the shape of a vortex generator on the stability of vortex generation in a vortex flowmeter. The article presents a numerical analysis of the influence of selected surfaces of the vortex shedder on the parameters of the vortex flowmeter. In order to determine the influence of the shape of the vortex shedder on the type of generated vortices, simulations were carried out for various flow velocities. Numerical calculations were experimentally verified for a cylinder-shaped vortex shedder. The experimental tests consist in determining the velocity field behind the vortex shedder. For this purpose, a proprietary method of determining local liquid velocities and the visualization of local vortices were used. On the basis of the conducted research, the influence of the shape of the vortex shedder on the width of the von Karman vortex street was determined and the optimal longitudinal distance from the shedder was determined in which it is most useful to measure the frequency of the vortices. This place ensures the stability of the frequency of the generated vortices.

## 1. Introduction

One of the solutions used to measure the velocity of liquids are vortex flow meters. These flow meters are used to measure the velocity of liquids of various densities and viscosities. Their advantage is that the measurement result is not significantly influenced by solid particles carried with the liquid. They can therefore be used to measure contaminated liquids. A second advantage is the very low pressure drop across the measuring device. Their disadvantage is the relatively small measuring range [1].

The principle of operation of the vortex flowmeter is to create regular vortices behind the generator placed in the flowing liquid. The frequency of the vortices is a function of the flow velocity. The vortex generator can have different shapes. This shape has a direct impact on the sensitivity and measuring range of the flowmeter. The frequency of the resulting vortices is a measure of the flow velocity, which is expressed by the Strouhal number [2]:(1)$$St=\frac{f\cdot d}{v},$$
where: *St*—Strouhal number, *f*—frequency of vortices, *d*—characteristic dimension, *v*—flow velocity.

The Strouhal number is constant regardless of the velocity of the liquid that passes the vortex generator. The advantage of this solution is that the frequency of the generated von Karman vortices does not depend on the rheological properties of the fluid, such as viscosity or density. As a result, the indications of the vortex flowmeter do not depend on the temperature and type of fluid.

It should be noted, however, that the von Karman vortex path arises for the turbulen flow. Thus, the Reynolds number should be greater than 5000 [3]. Therefore, the size of the vortex shedder bar should be adapted to the rheological properties of the liquid and the measuring range. The rheological properties of the fluid depend on the type of fluid, temperature and its physicochemical composition. Moreover, the presence of a significant amount of solid particles in a liquid significantly changes its rheological properties [4]. When selecting the diameter of the vortex shedder, the value of the Reynolds number calculated from the formula should be considered.
(2)$$Re=\frac{v\cdot d\rho }{\mu },$$
where: *ρ*—density of the fluid, *μ*—dynamic viscosity of the fluid.

In this work, tests were carried out for pure water at a temperature of 20 °C in the speed range from 0.5 to 2 m/s. Assuming a vortex generator size of 20 mm, Reynolds numbers are between 10,000 and 40,000. Thus, this is sufficient to produce a von Karman street.

Figure 1 shows a typical design of a flowmeter [5]. It consists of a vortex generator, which is mounted in the flow space. It causes vortices to form behind it, creating the characteristic von Karman vortex path. In order to measure the frequency of the vortices that arise, a pressure detector is mounted at a certain distance. The simplest solution is a pressure sensor with high measurement dynamics. The frequency of the vortices is measured electronically.

The shape of the vortex sheder has a significant impact on the parameters and metrological properties of the flow meter [6]. Its shape should ensure the formation of regular vortices in the largest possible measuring range. The most common shape is a cylinder. There are many works on this subject in the literature [1,7]. Classic designs of vortex flow meters are also used in two-phase flows [8]. The research in this paper will be compared to a cylinder-shaped shedder bar. The main disadvantage of the cylindrical shape is its very narrow measuring range [9]. Therefore, there is a need to look for other solutions [2]. There are many works in the literature which describe cylindrical shedder bar [1,7,10]. In this paper, this shape will be used as a reference in the assessment of the influence of selected shedder surfaces on the von Karman vortex path. The main disadvantage of the cylindrical shape is its narrow measuring range due to the stability of the generated vortices [7]. As the size and shape of the vortex generator has a large impact on the correct operation of the vortex flowmeter [2,11], the search for new solutions is still justified [9,12], despite the fact that many shapes of vortex generators are known.

Many vortex shedder bars solutions are used in commercial flow meters. Many vortex generator solutions are used in commercial flow meters. Despite so many solutions, no universal shape of the vortex shedder bar has been developed. This study is devoted to the analysis of the influence of selected surfaces of the vortex shedder bar on selected parameters of the vortex flowmeter. In the first part, numerical tests were carried out, the aim of which was to distinguish significant surfaces of the vortex generator, which may have a significant impact on the metrological parameters of the flow meter. Then, the selected shapes were tested experimentally.

A cylinder with a diameter of *d* = 20 mm was adopted as the reference shape (Figure 2). Eight surfaces were distinguished in the cross-section. Four of them form the rake face and the remaining four form the trailing face.

This solution can be used to measure the flow of clean and contaminated fluids with solid particles. Measurement error of the current value of the stream 0.5 ÷ 1% for liquids and approx. 2% for gases. The measurement error is significantly influenced by the regularity of the vortices formed. The complex nature of the resulting chips provides a lot of information about the process of their formation. This information can be processed on the basis of data fusing process [13].

In order to determine the influence of individual surfaces on the von Karman swirl pathway formed behind the vortex shedder bar. The main analysis was the influence of individual surfaces on the stability of the generated vortices. The results of the calculations were compared to the calculations for the cylinder-shaped generator. The research was carried out in such a way that selected surfaces from 1 to 8 were modified (Figure 2) and then numerical calculations were carried out for the same conditions as for the cylinder. Shapes with a modified rake face were proposed (Figure 3a). The second modification is the changes in the runoff surface (Figure 3b). The same characteristic dimension d = 20 mm was kept in all modifications. In addition, the effect of extending the dimension of the sherdder bar in the direction of flow motion was tested (Figure 3c). The elongation was 6 mm (W1) and 12 mm (W2). In addition, it was checked what effect is the use of a slot with a radius of 3 mm (W3) and a slot with a radius of 6 mm (W4) on the lower and upper surfaces of the shedder bar.

The aim of the work is to present the influence of changes in the shape of the rake and runoff surface in the vortex shedder bar on the stability of the vortices. The work carried out numerical and experimental tests on the basis of which the parameters for various shapes of vortex shedder bar were determined. 

## 2. Numerical Method 

Many mathematical models are evaluated to simulate flow velocity. The most popular models for modeling complex flow problems are numerical methods CFD (Computational Fluid Dynamics). These models are mainly based on Reynolds theory. It is assumed that fluid motion can be defined by two parameters such as average velocity and fluctuation velocity. As a result, the Navier–Stokes equation describing the motion of a fluid additionally contains a turbulence stress tensor [12]. This causes additional variables to appear in the equation. On correctness of the numerical results affects using suitable turbulence model and preparation right preprocessing parameters. To some of the most important aspects include the number of mesh elements and the areas of its density. Conducted research purpose the selection of the optimal turbulence model and to determine the minimum number of the mesh elements.

The appearance of additional variables in the Reynolds equations requires supplementing the matamatic model with additional equations. In practice, they are most often known as k-ε or k-ω turbulence models [12,14]. Each of these models has certain limitations. They make each of them reflect the real movement of fluids better or worse. In the first part of the work, research was carried out to identify the model that best reproduces the real shape of the von Karman vortex path.

The k-ε model takes into account the relationship between the vorticity and the kinetic energy of turbulence [14]. This model models well turbulence in free flow and in shear layers. It is characterized by low sensitivity to boundary conditions, especially to the distribution of liquid velocity at the inlet to the channel. This is a desirable feature due to the fact that these quantities are often not exactly known in practical calculations. Meanwhile, the k-ω model takes into account both the kinetic energy of turbulence and the speed of energy dissipation in its solution. In the k-ω model, like in k-ε, k will express the kinetic energy of turbulence and ω is the ratio of the energy dissipation speed ε to the kinetic energy k. The k-ω model, on the other hand, models the turbulent flow in the boundary layer much better, while it is very sensitive to the values of turbulent quantities in free flow.

It was found that you can combine the desired features of both models by combining them into a single model, in ANSYS Fluent it was named k-ω SST. The k-ω SST model was used for further numerical research. This model is a good representation of the kinetic energy of the vortices forming.

The appropriate selection of the finite element mesh has a significant impact on the simulation results [15]. The structural computational mesh was generated for rectangular elements of various shapes and sizes. The mesh is refined in the immediate vicinity of the vortex shedder bar (Figure 4). For the purposes of the research, a calculation grid was prepared for a rectangular duct with a length of L1 = 900 mm and a width of L2 = 600 mm. A cylinder with a diameter of 20 mm was placed in the middle of the channel. The width of the channel is so large that the influence of the side walls of the channel on the process of vortex formation can be neglected.

Figure 5 presents the results of numerical simulation of the number of mesh of finite 400,848 nodes. The k-ω SST model for a cylinder-shaped of shedder bar is a good representation of the von Karman street, so it can be used in the study of shedder bars of other shapes.

## 3. Computational Results

In order to investigate the influence of the shape of the vortex generator on the stability of the generated vortices, simulations were carried out for various flow velocities (0.5 m/s, 1 m/s, 2 m/s). Numerical calculations were performed with the ANSYS Fluent software. The calculations were carried out on a computing server consisting of two computers. The first had 24 cores and the second had 20 cores In numerical calculations, the Turbulence k-ST SST model was used for a two-dimensional velocity field and a non-stationary flow. The boundary conditions were the assumption that there is a uniform velocity distribution in the inflow channel and that the liquid flowing in the channel is water. 

The calculation results were compared with the results for the cylinder-shaped generator. For the digital genaric vortex, regular and stable vortices were obtained for a liquid flow velocity of 0.5 m/s, 1 m/s and 2 m/s. The results of the numerical calculations are presented in Table 1. The frequency of the generated vortices was determined on the basis of how the pressure changed at the selected point on the basis of successive time steps. The calculation results for other shapes of shedder bars will be compared to the values presented in Table 1.

The main objective of the numerical research was to determine the effect of individual surfaces of the shedder bar on the frequency, pressure and stability of vortex formation. These are the parameters that have a direct impact on the proper operation of the vortex flowmeter. The range of changes in the frequency of the generated vortices affects the sensitivity of the flowmeter.

Figure 6 shows the numerical results of comparing the values of frequency changes of the vortices formed for various modifications of the vortex generator. The designations of the shapes correspond to the markings in Figure 2. The values in the diagram represent the ratio of the frequency of vortices generated by individual shapes to the reference frequency f_0_ (Table 1).

The more the f/f_0_ ratio differs from 1, the greater the impact on the sensitivity of the flowmeter will be the modification of the surface areas consistent with the markings in Figure 6. As it results from Figure 6a, the greatest impact on the sensitivity of the vortex flowmeter is the modification of areas 1 and 3 together with areas 2 and 4. In this type, the greatest variations are obtained for different flow velocity, but the variations are non-linear. Modifying only areas 3 and 4 or 1 and 2 does not bring such good results. The modifications to the P3 and P4 deserve special attention here

In the case of the runoff surface modification, a significant improvement in the flowmeter sensitivity is obtained for the shapes T2 and T7. These shapes have a sharp edge, which makes it easier to detach the vortices.

Modification of the upper and lower flow surfaces of the vortex shedder bar (shapes in Figure 6c) mainly increases the frequency of the generated vortices, but does not significantly affect the sensitivity of the flowmeter. Only the modification of W3 has a visible effect on the sensitivity.

The second important parameter influencing the operation of the vortex flowmeter is the value of pressure changes of the generated vortices. The greater the pressure change of the generated vortices, the greater the signal-to-noise ratio in the measurement signal. Figure 7 shows the results of the numerical simulation. These results show the impact of modification of the rake and trailing surface on pressure changes generated by vortices formed behind individual shedder bars.

The greatest changes for different fluid streams are obtained for modifications P3 and P4 (Figure 7a) and for modifications T1 and T2 (Figure 7b). The least effect is obtained for the T7 modification. On the other hand, for the W1 modification, the changes increase with small flow rate, while for larger flows they remain at the same level. Modification of W3 lowers the level by approximately a constant amount.

Based on the research, it was found that the modification of the rake surface, consisting in increasing the drag coefficient, causes an increase in the pressure difference behind the vortex generator. This improves the signal-to-noise properties of the flowmeter, but the processing characteristics become more non-linear. This will make a difference at higher fluid flow rates.

The stability of the von Karman street has a great influence on the correct work of the vortex flowmeter. Further research was carried out to determine the impact of changes in the shape of rake surface on the stability of vortex street. Modifications of the shape of shedder bar P3, P4, T1, T2, T7, W1 and W3 were selected for further numerical tests.

The results of the analysis for the shapes P3 and P4 are shown in Figure 8. Modification of the rake surface destabilizes the vortex street in a wide range of velocity of the tested fluid. The destabilization of the vortex street is already noticeable for a very low velocities of flow velocity *v* = 0.5 m/s (Figure 8a,b). Even a gentle flattening of the rake face of the vortex sedder bar causes considerable disturbance to the formation of the vortex street. Modification P4 may be appropriate for very low flow rates as it generates the greatest amplification of the pressure amplitude downstream of the vortex generator. Moreover, it extends the measuring range. The analysis shows that in order to obtain good stability of the generated vortices, it is best if the rake surface is as streamlined as possible. It will be a good solution especially for measuring high velocities of flowing fluid.

Analyzing the obtained results for the shapes in which the runoff surfaces T1 and T2 were modified (Figure 9a,b), it was found that the modification of these surfaces did not destabilize the stream. The reason is that the stream of flowing liquid mainly breaks off from the top and bottom surfaces of the shedder bar. These modifications advantageously affect the amplitude of pressure changes of the resulting vortices.

The T7 modification (Figure 9e,f) has an even smaller impact on the stability of the von Karman swirl path. Since Modification T7 has little effect on improving the value of the pressure amplitude of the resulting vortices, it is advisable to pay special attention only to the modification T2.

The streamlined shape of the rake surface has a positive effect on the stabilization of the vortex street for high flow velocities. Sharp ending of the trailing surface does not destabilize the vortex street, but improves the sensitivity of the vortex flowmeter. Thus, it has a positive effect on improving the parameters of the vortex flowmeter. To increase. The modification of W3 does not significantly change the parameters of the vortex flowmeter; however, it linearizes its characteristics. Based on this analysis, for experimental research, it was decided to develop the shapes of vortex generators which constitute a kind of compilation of the modifications presented in Figure 2. Figure 10 shows the proposed shapes of vortex shedder bars that have been subjected to experimental tests.

## 4. Test Stand for Experimental Research

In order to verify the numerical calculations and determine the influence of the dimensions and shapes of the vortex shedder bar on the characteristic dimensions related to the construction of vortex flow meters, a test stand was built and a measurement method was developed to enable the visualization of vortex’s arising behind the shedder. 

In order to visualize the vortex path, the marker method was used [16]. This method enables the measurement of the local velocity of fluid movement and the tracing of eddies that occur. The selection of tags has a great influence on the results of the visualization. They should be of an appropriate size and the material from which they are made should have a density similar to that of the fluid. These markers should not significantly interfere with fluid movement. Water with polyamide markers with a density of 940 kg/m^3^ and granulation from 20 μm to 1 mm was used for the experimental tests. The markers had a good reflective surface.

The measurement method presented in this article is similar to the PIV. However, it differs in the method of calculating the local speed value. This method consists in registering the trajectory of the movement of markers moving with the liquid. Compared to the PIV method, the tags have much larger dimensions. This kind low-cost DPIV set-ups, using free PIV software and high-speed cameras instead of the classical double exposure cameras, have been successfully employed in PIV and liquid-granular PIV applications [17,18,19]. The basis for calculating the speed of movement of the tag is the analysis of its movement based on the image of only one image of the fluid flow. This method does not require image correlation of at least two consecutive photos, as is the case with the PIV method. The image of a moving particle is recorded with a long exposure. The exposure time is so long that the marker in the image is visible in the form of a longitudinal streak with a shape corresponding to the marker movement trajectory.

The local liquid velocity is determined on the basis of the traces left by the markers (Figure 11). After hitting the marker, the light that shines through the measurement section is reflected from its surface. In cameos it is visible as a bright spot. When a long exposure time is used, the trace of the moving marker will be visible as a bright line.

The length of the trace left by the marker in the image is the basis for calculating the local velocity and direction of movement of the fluid. The local speed is calculated from the formula:(3)$$v=\frac{k\cdot p}{t},$$
where: *v*—velocity of liquid flow, *k*—scale coefficient, *p*—number of pixels, *t*—exposition time.

On the basis of Equation (3), the instantaneous values of the fluid movement are calculated. This method determines the instantaneous velocities at various points in the flow and at various distances from the vortex generator. In the further part of the work, the velocities of the liquids refer to the values averaged from the instantaneous values in the entire observed cross-section.

The longer the exposure time and the higher the speed of liquid movement, the longer the streak recorded by the moving marker will be (Figure 12). Long marks are advantageous for the accuracy of the measurement. However, due to the fact that the markers move in three planes for long tracks, it often happens that the marker leaves the space illuminated by the laser. Therefore, there is an optimal exposure time for the selected interval of fluid movement velocity.

For the purpose of the research, the optimal exposure times for the velocity of liquid movement were determined in the measurement channel. The influence of exposure time on the uncertainty of measurement of the velocity of liquid movement was investigated. The uncertainty increases both when the exposure time is too long and also when it is too short. In the case of short exposure times, the measurement error is mainly affected by the resolution of the recorded images. This is due to the formation of very short trace that are left by mud flakes. Since the camera has limited possibilities of changing the resolution, the measurement uncertainty is optimized by means of a properly selected exposure time.

Based on experimental studies, the recommended exposure times for selected velocities of fluid movement were determined. A number of measurements were made for different exposure times for markers moving in the speed range from 0,01 m/s to 0,65 m/s. The tests were carried out for the following exposure times 1/5 s, 1/10 s, 1/20 s, 1/30 s and 1/50 s. The uncertainty for a Type A was calculated based on the formula [20]:(4)$$U=\frac{s}{\sqrt{N}}=\sqrt{\frac{\sum_{n=1}^{N}{\left(v_{i}-v_{s}\right)}^{2}}{N\left(N-1\right)}},$$
where: *s*—estimated standard deviation, *v_i_*—flow velocity, *v_s_*—mean flow velocity, *N*—number of measurements.

The mean flow velocity is calculated on equation [20]: (5)$$v_{s}=\frac{1}{N}\overset{N}{\underset{n=1}{\sum }}v_{i}$$

Figure 13 shows the results of research on the influence of exposure time on the uncertainty of the measurement of the speed of movement. To make the presented results easier to read, the uncertainty values were expressed as a percentage. The percentages represent the ratio of uncertainty to the average velocity of fluid movement. Calculated based on the formula
(6)$$U\left[\%\right]=\frac{U}{v_{s}}\cdot 100\%,$$

As it results from the measurements, exposure times below 1/10 s are recommended to be used for liquid movement speeds below 100 mm/s. However, for liquid movement speeds below 500 mm/s, it is recommended to use exposure times shorter than 1/30 s.

The accuracy of the speed measurement is significantly influenced by the appropriate selection of the exposure time. In order to select the optimal exposure time, studies were carried out on the influence of exposure time on the error of local velocity measurement depending on the liquid velocity [21]. Based on experimental studies, exposure times for different velocities of fluid movement have been proposed (Table 2). By analyzing the results in Table 2, it can be concluded that there are at least two recommended exposure times for most measuring ranges. While there is some flexibility in the choice of exposure times, there are some threshold values that should not be exceeded due to the measurement uncertainty.

As it results from the conducted analysis, the inaccuracies in measuring the velocity of the liquid when selecting the appropriate exposure time are in the order of a few percent. These speeds were compared with those calculated on the basis of the measurement of the liquid stream flowing in the channel. The liquid flow was measured with a standardized measuring orifice. When comparing the results, a similar accuracy was obtained and the discrepancy of the results did not exceed 5%. Such accuracy was found to be sufficient for the purposes of this study. 

From the consecutive photos it is possible to follow the motion of the vortex path (Figure 14). By analyzing consecutive photos, it is possible to determine when the maximum turbulence occurs at a selected distance from the vortex shedder. For example, in Figure 14a there is an observation point in which the maximum of the occurring vortex is visible for 66 frames of the image. Then, for the frame 76, this maximum is formed on the other side of the horizontal axis (Figure 14b). Again, the vortex maximums for the selected viewpoint will appear in the 86 frame of the image (Figure 14c). On this basis, the number of frames between successive vortex maxima is calculated. Since the camera captures the image at a constant frame rate, the duration of the vortices can be calculated from the frame rate. The frequency of the resulting vortices was calculated on the basis of the period.
(7)$$f=\frac{1}{T}=\frac{N_{F}}{\left(t_{2}-t_{2}\right)}$$
where: *t*_2_*, t*_0_—number of the image frame in which the vortex maxima appear at the observed point, *N_F_*—number of frames of the recorded image per second.

In order to conduct experimental research, a measuring stand was designed and built (Figure 15). The resulting vortices are observed in a transparent channel (1) with a rectangular cross-section and dimensions of 50.0 × 250 mm. The channel is entirely made of acrylic glass. In its upper part, a cover is mounted, thanks to which it is possible to mount various vortex generators in the measuring space. The water in the channel is in a closed circuit and is stored in a tank (2) with a capacity of 1 m^3^. The liquid is pumped by a pump (3) with a capacity of 17 m^3^/h and a maximum head of 18 m. The pump is driven by a 3 kW electric motor. The water flow in the channel is regulated by the throttle valve (5). The valve (4) is used to regulate the pressure in the duct. In order to obtain the same local velocities at the inlet to the measuring channel, a flow straightener (6) was used. The straightener is shaped to condition the flow by increasing the flow velocity along the channel walls and reducing turbulence. In the middle of its length, there is an additional bundle-pipe straightener. Above the measuring section, the Sony ILCE-7SM2 digital camera was installed, which recorded the movement of the markers. The camera allows you to record movies with Full HD resolution and a maximum speed of 120 fps. The camera allows you to set the exposure time in the range from 1/8000 s to 30 s.

The experimental tests were conducted over a length of 450 mm. The measuring section was illuminated in a horizontal plane from two directions with the use of line lasers.(Figure 16) The lasers had a peak power of 1 W and the emitted wavelength was 520 nm. They were semiconductor lasers equipped with a collimator generating a line with a scattering lens. The use of lighting from both sides significantly improved the projection of the particles, as it reduced the appearance of a dark shadow behind the illuminated particle. The lasers were installed along the sides of the channel in such a way that the liquid film with a thickness of 1mm was illuminated in the middle of the channel height.

## 5. Validation of the Results of Numerical Calculations with the Experiment Results

A series of experimental studies were carried out to investigate the influence of the shape of the vortex shedder bar of the assumptions made for the numerical calculations, the results of the numerical calculations were compared with the results of the experiment. The basic value measured in vortex flow meters is the vortex frequency; therefore, the basic validation criterion was the change of the frequency of the generated vortices from the flow velocity. Figure 17 shows the results for a cylinder-shaped shedder.

Figure 17 shows the dependence of the frequency of the generated vortices as a function of the velocity of fluid. Particular relationships were determined for various diameters of cylinder. The red dotted line represents the results obtained from the numerical simulation for a cylinder with a diameter of 20 mm. 

The continuous lines approximated the results of experimental measurements for various dimensions of the vortex generator. By comparing the obtained results, a satisfactory convergence of the results was obtained. The results of the numerical simulation show a tendency to overestimate the frequency of the generated vortices. The reason for this may be the approximations resulting from the turbulence model used. This inaccuracy can be corrected by introducing an appropriate value of the constant component. Another important parameter is the angle of the solid line. This angle is the greater the smaller the size of the generator. The greater the angle of inclination, the greater the sensitivity of the flowmeter. However, the use of small dimensions of the generator limits the range of applicability as it is required that the Reynolds number should not be less than 5000 for the entire measuring range. Analyzing the slope of the dashed theoretical line, it is greater than it would appear from the experimental measurements. The reason is that the numerical tests do not consider the influence of the side walls. 

Since the numerical tests described in this article were aimed at determining how individual surfaces of the vortex shedder bar affect the parameters of the flowmeter and the results were related to the cylinder-shaped generator, the obtained convergence of the numerical results with the experiment results can be considered sufficient. The basic criterion for the selection of a vortex generator is the linearity of the frequency velocity dependence. As can be seen from Figure 1, the frequency of the resulting vortices depends linearly on the velocity of the floating liquid. This tendency is consistent with the theoretical relationship expressed by the formula (1). On this basis, the correctness of the assumptions resulting from the theoretical analysis is confirmed, which assumptions were used to develop specific shapes of vortex generators, which were further tested experimentally.

The range of frequency changes strongly depends on the characteristic size of the generator. The characteristic dimension also affects the slope of the characteristic and the value of the zero point. The zero factor k is in practice only determined during the first calibration and does not change throughout the service life. The value of the flow velocity is calculated from the relationship where the ratio *d/St* is the slope of the linear dependence *v = f (f)*:(8)$$v=f\frac{d}{St}+k$$
where: *St*—Strouhal number, *f*—frequency of generated vortices, *d*—characteristic dimension of the obstacle, *k*—zero factor

Table 3 presents functions approximating the results of the experimental data presented in Figure 17. Based on these functions, it is possible to calculate the values of the Strouhal number for a shedder bar and the value of the k coefficient.

Figure 18 shows the dependence of the value of the Strouhal number on the value of the characteristic dimension. The linearity of this relationship proves that the measurements were correctly carried out. Similar compliance with the theoretical assumptions is demonstrated by the linear nature of the d/St slope coefficient. As shown by the performed measurements, the coefficient of zero k is a non-linear dependence on the characteristic dimension. Hence, it is necessary to determine this coefficient individually for each type, shape or size change of the vortex shedder bar. For the generator used in the experiment, the translation function of calculating the *k*-factor depending on the diameter of the generator is: (9)k=0.18log(d−6.5)−0.24

Based on the analysis of the image of the von Karman vortex street, it is possible to determine two basic characteristic dimensions (Figure 19). The most important dimensions are the length of the vortex zone *l* and the deviation from the axis *h*. These dimensions depend on the size and shape of the shedder bar. These dimensions are important as they determine the optimal location for install a vortex detector. In addition, the dimensional ratio *h*/*l* is a criterion for the stability of vortices. It is believed that it should be in the range from 0.15 to 0.45 [22]. Figure 19 shows an exemplary image of a von Karman street captured with a camera. The image shows the method of determining the parameters *l* and *h*. The distance *l* is determined from the canter of the shedder bar to the canter of the vortex image. The distance h is the distance from the flow axis to the edge of the vortex.

The next part of the research is to determine the relationship between the characteristic dimension of the vortex shedder bar *d* and the dimensions characterizing the von Karman vortex street. Moreover, it was determined how the change of the shape of the vortex generator influences the change of the characteristic dimensions of the vortex street (Figure 20). 

These equations can be used to calculate the dimensions characteristic for paths resulting from the flow of vortex shedder bar of other shapes.

Subsequent research consisted in determining the effect of changing the shape of the vortex shedder on the characteristic dimensions of the von Karman vortex street (Table 4). The *d* dimension of all tested shedder bars was the same and amounted to d = 30 mm. The results were compared with a cylindrical shedder with a diameter of d = 30 mm. For this shedder, the characteristic dimensions of the von Karman vortex street were *h_o_* = 18.2 mm and *l_o_* = 51.5 mm. The results presented in Table 3 are averaged values for the liquid velocity from 0.3 m/s to 1 m/s.

As it results from the research, changing the shape of the rake surface to a more streamlined one shortens the non-vortex zone behind the shedder (E1). In turn, the introduction of a slot on the lower and upper surface of the generator causes a slight elongation of this zone (E2). The introduction of the slot, in turn, significantly reduces the distance *h*. Modification of the generator runoff surface to be more streamlined causes that the distance *h* is greater (E5). Good stability of the swirl street while maintaining favorable values of *h* and *l* is obtained for the E 4 shape of the vortex shedder bar. The shape E3 reduces the characteristic dimensions of the vortex street by 7% compared to the cyclic shedder, but introduces the greatest instability of the resulting vortices (*h*/*l* = 0.4). The introduction of a sharp rake face usually increases the distance *h*.

## 6. Conclusions

The paper presents results of numerical studies on the impact of changes in surface of flow around vortex shedder on the parameters of von Karman vortex street. Based on the numerical results, several shapes of vortex shedders were selected, which improve the metrological parameters of the vortex flow meters (Figure 3). Selected shapes were subjected to experimental tests on a test stand. In this way, the results of numerical tests were verified and it was found that the most useful solution from the proposed solutions of shedders bars is the E4 shape for metrological reasons. 

For the purposes of the study, a method of visualization of the von Karman vortex street was developed. The measurement method presented in the work can be used to determine the velocity of liquid flow in the visualization processes. The only limitation is the use of trans-parent liquids. Due to the fact that many transparent liquids exist in technical conditions, its range of use is very large. The presented approach offers the possibility to determine liquid flow velocity with a precision in the range of up to a few per cent in relation to the velocity of the liquid flow. The uncertainty of such measurements is considerably relative to the liquid velocity. An adequate selection of the exposition time forms a vital aspect in ensuring the required accuracy of the measurements; yet it is relative to the velocity of the analyzed phenomenon. The analysis described in this paper provides valuable insights into the selection of exposition times in studies involving visualization of the liquid flow.

The measurement method can be used to visualization the velocity of liquid flow in the other processes. The only limitation is the use of trans-parent liquids. Due to the fact that many transparent liquids exist in technical conditions, its range of use is very large. The presented approach offers the possibility to determine liquid flow velocity with a precision in the range of up to a few per cent in relation to the velocity of the liquid flow. This paper presents into the selection of exposition times in studies involving visualization of the liquid flow, which has a great influence on the accuracy of the measurement.

On the basis of the observations, it was found that the introduction of the fracture improves the stabilization of the vortex street. Jet is beneficial in terms of improving the regularity of vortex formation. When the rake surface is more streamlined, it helps to stabilize the stream at higher speeds of movement. W guiding the sharp edges on the trailing surface causes the distance h to be greater, but improves the widening of the lower range of vortex formation. The introduction of the concave rake face destabilizes the vortex street, which is a disadvantage.

## Figures and Tables

**Figure 1 sensors-21-04697-f001:**
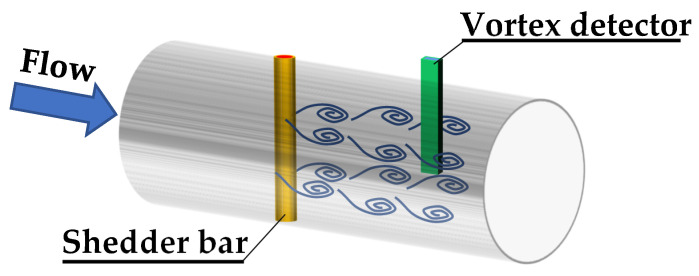
Vortex flowmeter.

**Figure 2 sensors-21-04697-f002:**
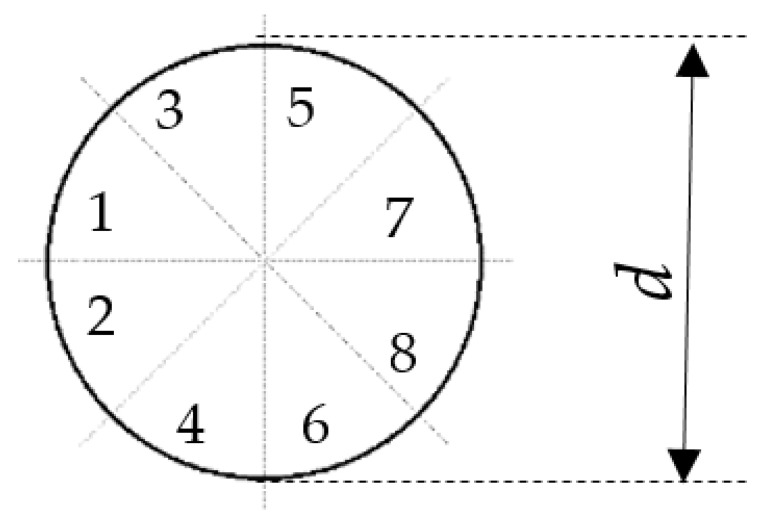
The basic vortex shedder shape.

**Figure 3 sensors-21-04697-f003:**
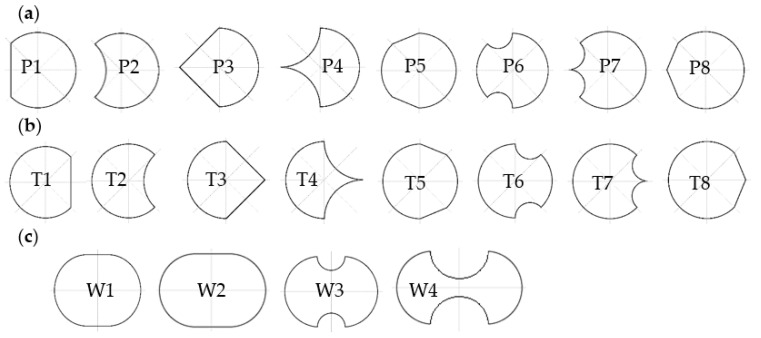
Shapes resulting from (**a**) rake surface modification (**b**) runoff surface modification (**c**) oval.

**Figure 4 sensors-21-04697-f004:**
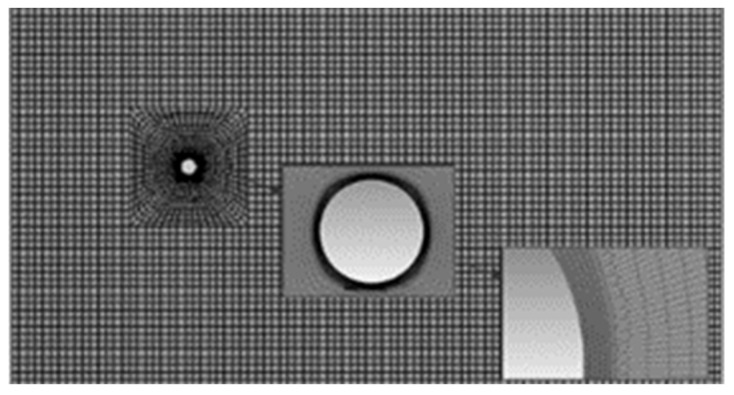
Computational mesh.

**Figure 5 sensors-21-04697-f005:**
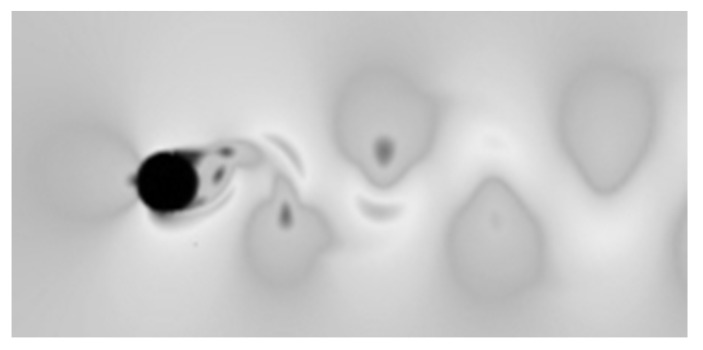
The results of numerical calculations for the k-ω model used in the further part of the work.

**Figure 6 sensors-21-04697-f006:**
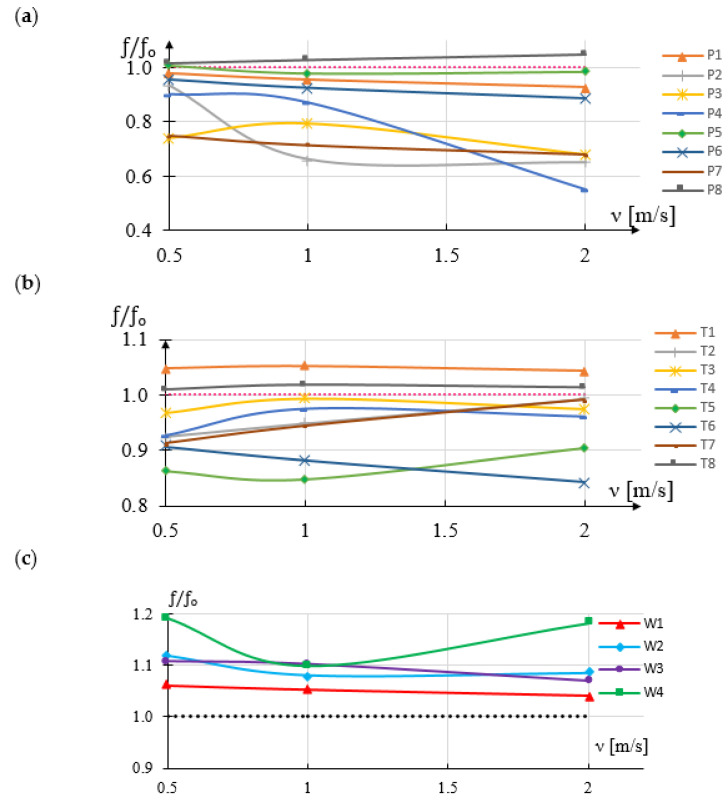
Shapes resulting from (**a**) rake surface modification (**b**) runoff surface modification (**c**) oval.

**Figure 7 sensors-21-04697-f007:**
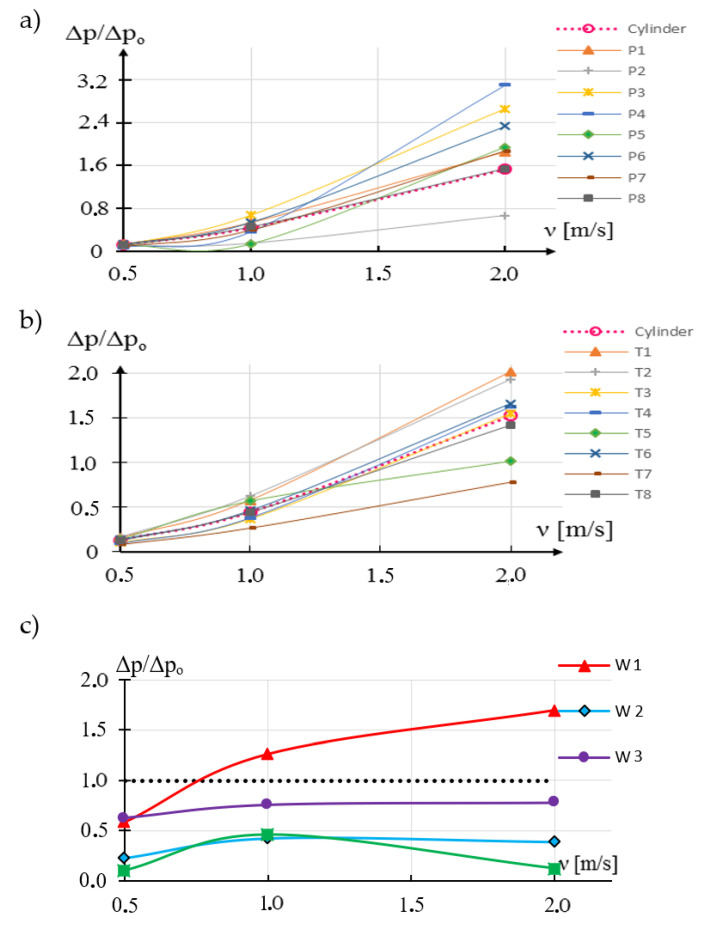
Shapes resulting from (**a**) rake surface modification (**b**) runoff surface modification (**c**) oval.

**Figure 8 sensors-21-04697-f008:**
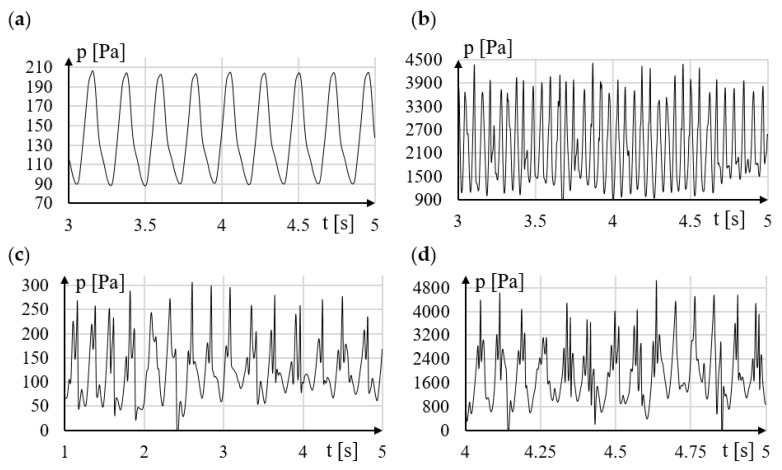
The influence of 1, 2, 3 and 4 surfaces modification on regularity of generated vortices for (**a**) P3 with v = 0.5 m/s, (**b**) P3 with v = 2 m/s, (**c**) P4 with v = 0.5 m/s, (**d**) P4 with v = 2 m/s.

**Figure 9 sensors-21-04697-f009:**
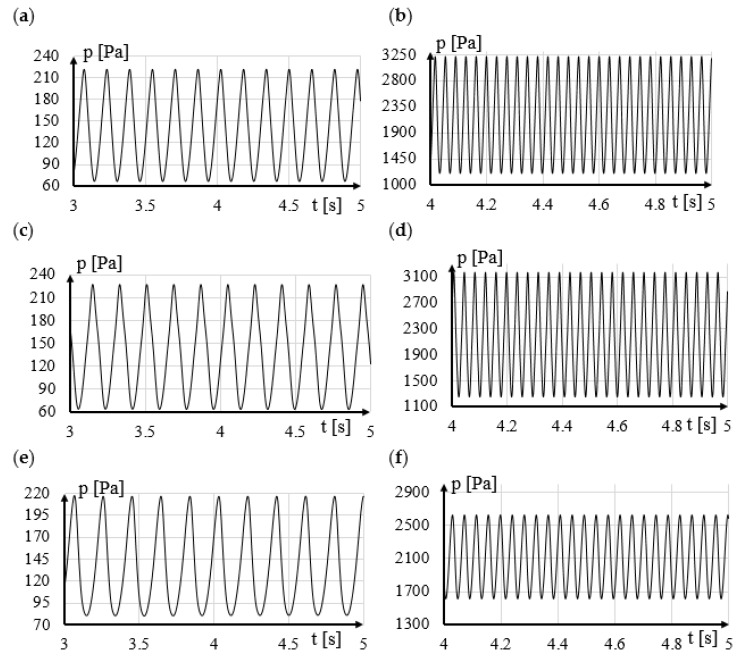
The influence of 5,6,7 and 8 surfaces modification on regularity of generated vortices for (**a**) T1 with v = 0.5 m/s, (**b**) T1 with v = 2 m/s, (**c**) T2 with v = 0.5 m/s, (**d**) T2 with v = 2 m/s, (**e**) T2 with v = 0.5 m/s, (**f**) T2 with v = 2 m/s.

**Figure 10 sensors-21-04697-f010:**
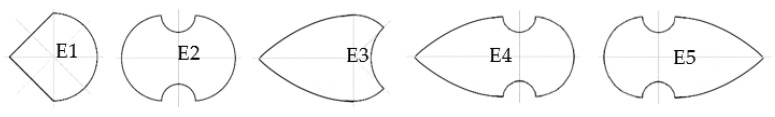
Experimental shapes of vortex shedder bars.

**Figure 11 sensors-21-04697-f011:**
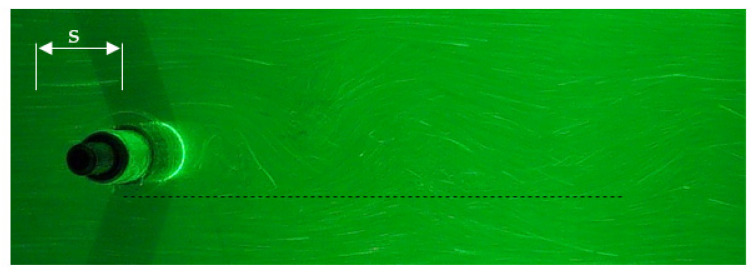
Image of tracer particle flow.

**Figure 12 sensors-21-04697-f012:**
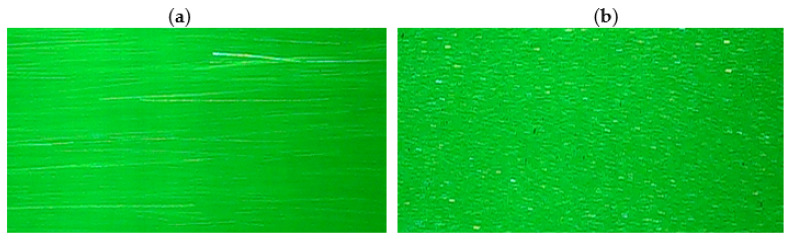
Sample images of registered trace markers for (**a**) exposure time 1/10 s of particle velocity 0.799 m/s, (**b**) exposure time 1/50 s of particle velocity 0.047 m/s.

**Figure 13 sensors-21-04697-f013:**
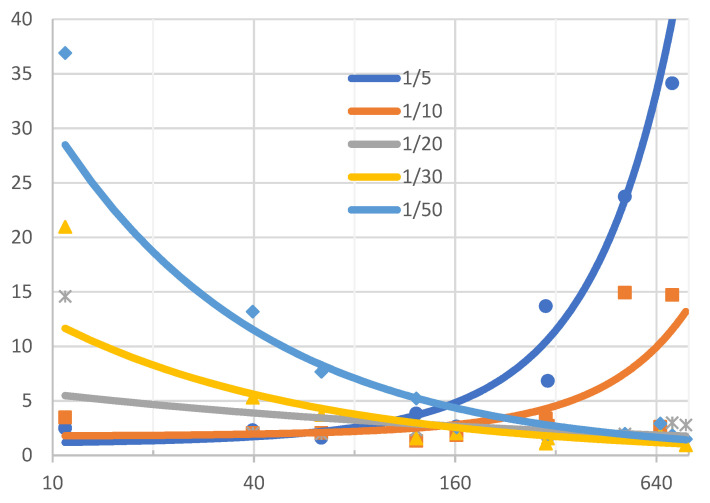
The results of the calculation of the measurement uncertainty for different exposure times and the speed of the marker movement.

**Figure 14 sensors-21-04697-f014:**
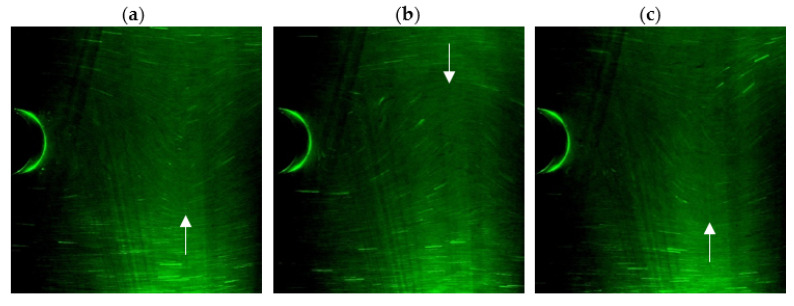
Sequence of consecutive photos (**a**) t_0_ = 66, (**b**) t_i_ =76, (**c**) t_2_ = 86.

**Figure 15 sensors-21-04697-f015:**
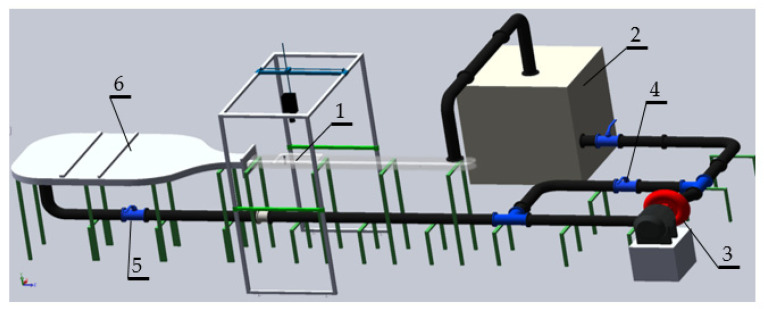
Experimental setup.

**Figure 16 sensors-21-04697-f016:**
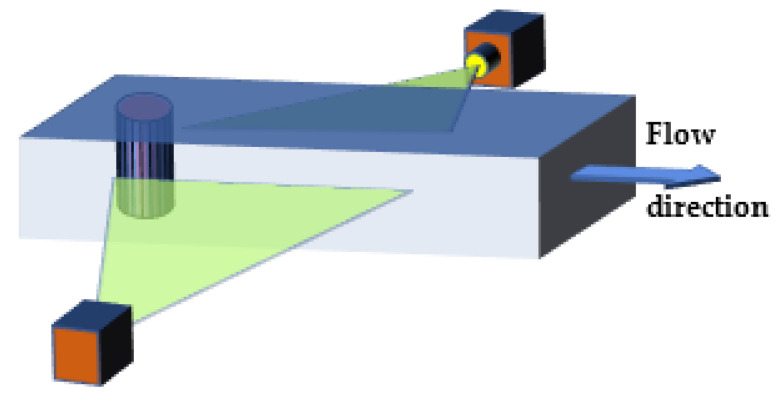
Experimental section.

**Figure 17 sensors-21-04697-f017:**
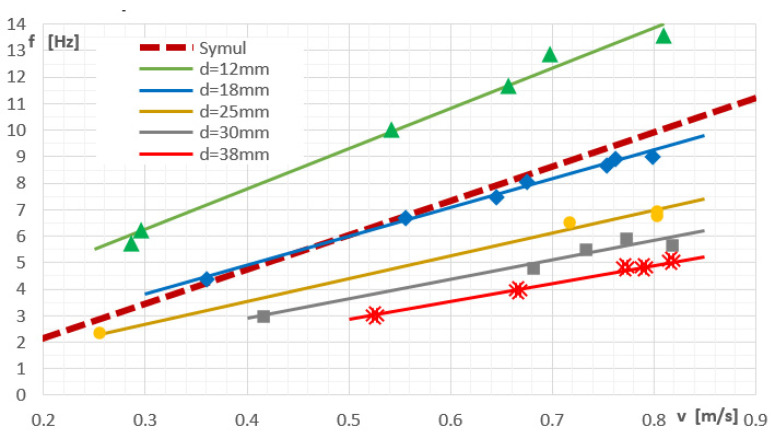
The dependence of the frequency of the generated vortices on the velocity of fluid movement.

**Figure 18 sensors-21-04697-f018:**
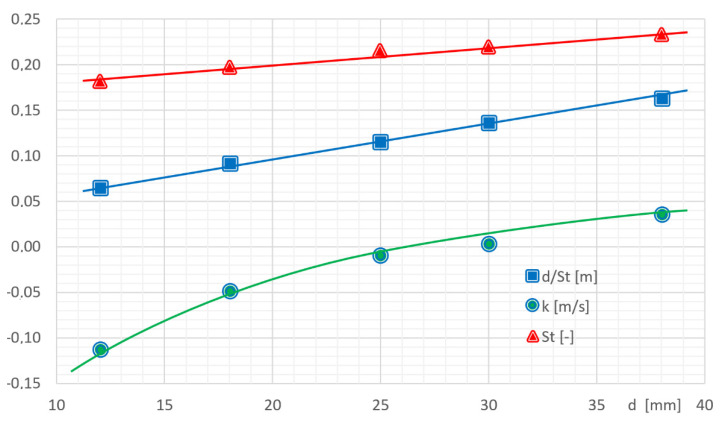
Influence of the characteristic dimension on the parameters of Equation (3).

**Figure 19 sensors-21-04697-f019:**
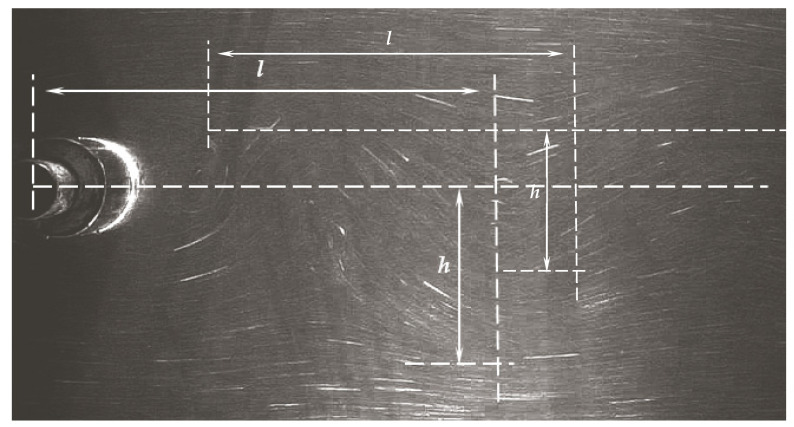
Schematic of geometry of von Karman vortex street.

**Figure 20 sensors-21-04697-f020:**
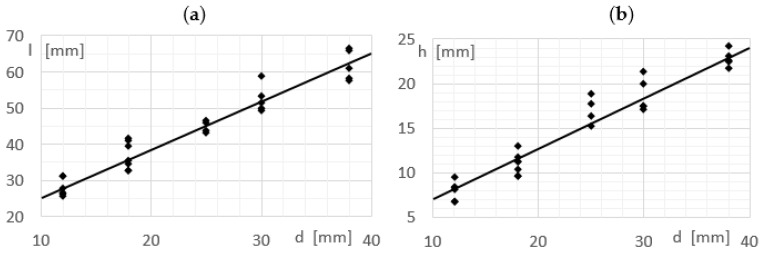
Influence of shedder size on (**a**) length of the vortex zone *l* (**b**) distance from axis *h*.

**Table 1 sensors-21-04697-t001:** Computational results for the vortex shedder from the Figure 2.

v [m/s]	f [Hz]	Δp [Pa]
0.5	6.01	128.11
1	12.44	442.51
2	26.32	1529.5

**Table 2 sensors-21-04697-t002:** Recommended exposition times depending on the liquid velocities [21].

Recommended Exposition Time [s]	Mean Velocity of Tracing Particles [m/s]
1/5	<0.1
1/10	<0.15
1/20	>0.1
1/30	>0.15
1/50	>0.4

**Table 3 sensors-21-04697-t003:** Mathematical functions approximating the results of the experimental.

d [mm]	Mathematical Function of Frequency
12	*f* = 0.0152 ∗ v + 1.7054
18	*f* = 0.0109 ∗ v + 0.5273
25	*f* = 0.0086 ∗ v + 0.0769
30	*f* = 0.0073 ∗ v − 0.0333
38	*f* = 0.0061 ∗ v − 0.2214

**Table 4 sensors-21-04697-t004:** Influence of changing the shape of the vortex shedder bar on the average dimensions of the vortex street.

Shedder Shape	*h* [mm]	*l* [mm]	*h*/*l*	*h*/*h_o_*	*l*/*l_o_*	*h*/*d*	*l*/*d*
E1	18.75	47.9	0.39	1.03	0.93	0.63	1.6
E2	10.38	52	0.2	0.57	1.01	0.35	1.73
E3	16.9	42.23	0.4	0.93	0.93	0.56	1.41
E4	10.37	49.44	0.2	0.57	0.96	0.34	1.65
E5	10.56	47.38	0.22	0.58	0.92	0.35	1.57

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
