# Peer review of "Analysis of the Influence of the Vortex Shedder Shape on the Metrological Properties of the Vortex Flow Meter"

_sensors, 2021, doi:10.3390/s21144697_

Round 1

Reviewer 1 Report

I noted that the manuscript has responses to the frenquency measuring method , however, Fig.14 needs more explanations, it looks like a carfully edited picture, compared to the images of karman vortex street using conventional methods , the shape of vortice looks strange; in addition, how are h and l determined?

Author Response

  1. The drawing has been changed. The h and l dimensions are marked in the example flow image from the camera.
  2. A comment was added in the text. 

Reviewer 2 Report

Dear Authors,

the manuscript has been significantly improved. I acknowledge the efforts made.  Please have a final English language and style check.  Label the axes of Fig. 13 with the physical quantity being plotted. Do not forget to include the units.

Regards

Author Response

  1. The units in the chart have been corrected.
  2. A linguistic proofreading was commissioned at the publishing house 

This manuscript is a resubmission of an earlier submission. The following is a list of the peer review reports and author responses from that submission.

Round 1

Reviewer 1 Report

the numerical method proposed in this manuscript could simulate the vortices behind a cylinder and determine their frequency & amplitude of pressure, which may have reference value for some readers,  however the authors only shown a visualization method for the velocity measurement, I don't know how they determine the vortex's  frequency and amplitude in experiment?

Reviewer 2 Report

The paper is not suitable for consideration because of the following reasons:

  1. Experimental technique is neither established nor validated; there is no information about the spatial and temporal output of the technique; authors compare instantaneous velocities to mean velocities to evaluate uncertainty, which is basic mistake.
  2. Numerical method neither properly described nor validated. Authors even present results from DNS, which would be very improbable to do with the computational resources available to them.
  3. Authors provide inadequate descriptions of both experimental and numerical techniques, methodologies, of the relevance of the problem, of the parameters (not even the Reynolds number is given), of the literature on the topic, etc. Authors seem to be uninformed of fundamental issues regarding experimentation and simulation as well as reporting of scientific results.
  4. The study concerns two-dimensional flow around a circular cylinder which has no relevance to the topic of sensors – vortex flow meters do not use circular cylinders as vortex shedders while the vortex shedders are placed in circular ducts.
  5. The paper is badly written with errors and inconsistencies. E.g. velocities are measured in mm. The text even contains irrelevant comments about “the time of the conference”. Have authors checked their manuscript?

Reviewer 3 Report

The manuscript reports a numerical/experimental study on vortex flow meters, with focus being given to the effects resulting from changes of size of the vortex generator. The numerical analysis is performed using the k-ω turbulence model, while a very simple trace-tracking method is proposed for experimentally estimating the flow velocity. The manuscript appears reasonably clear, while the small amount of details can be barely regarded as sufficient even for a short communication. The topic is certainly of interest for the readers of the journal Sensor, although the contribution is not particularly relevant. The use of English should be improved.

Major issues:

-from Section 1, I understand that the size of the vortex flowmeter investigated numerically is only 20mm. According to the scope of the paper, why did not the Authors numerically investigate also other sizes? Although this size of 20mm is suitable for clear water applications, it might be too small for solid-liquid mixtures. Since the Authors repeatedly claim that the measuring method can be used for both clear water and contaminated fluids with solid particles, they should briefly comment the possible limitations of the flowmeter size in these latter applications.

-in the Introduction (Section 1) and also at the beginning of Section 4, where the visualization methods
are introduced, some further literature support should be added. In particular, I suggest to add some key references about the application of vortex flowmeters to liquid-solid mixtures and contaminated fluids (as claimed as possible at lines 25-28 and 65).

-With reference to visualization methods such as PIV, at line 196 I suggest to mention that low-cost DPIV set-ups, using free PIV software and high-speed cameras instead of the classical double-exposure cameras, have been successfully employed in PIV and liquid-granular PIV applications:

-Sarno, L., Tai, Y. C., Carravetta, A., Martino, R., Papa, M. N., Kuo, C. Y. (2019). Challenges and improvements in applying a particle image velocimetry (PIV) approach to granular flows. J. Phys. Conf. Ser., 1249(1), 012011.

-Gollin, D., Brevis, W., Bowman, E. T., Shepley, P. (2017). Performance of PIV and PTV for granular flow measurements. Granul. Matter, 19(3), 42.

-Thielicke, W., Stamhuis, E. (2014). PIVlab–towards user-friendly, affordable and accurate digital particle image velocimetry in MATLAB. Journal of open research software, 2(1).

-The Authors propose to estimate the tracers trajectories by a single long-exposure photograph. In my opinion the reliability of this very simple method has not been sufficiently investigated and may be too low for the proposed application, especially considering that the fluctuation velocities are non-negligible in the turbulent regime. Could the Authors provide some brief comparison with other well-established measuring methods? How did the Authors deal with possible superposition of multiple traces in the same photograph? What is the recommended seeding density for this application? Would it be possible to introduce in the future a sub-pixel accuracy for an enhanced reliability of the method? I strongly suggest the Authors to elaborate and carefully discuss on these possible shortcomings in Sections 4-5.

-In Section 2, it is finally stated that “the k-ω model was found to be the optimal model for simulating the von 131 Karman vortex path”. In the previous discussion, further insights about the model could be given, concerning related advantages and disadvantages.

-In Section 4, please give more details about the laser devices. Are these lasers equipped with a standard cylindrical lens to get a planar sheet of radiation? Why was it necessary to employ two lasers instead of just one? Where is "the plane halfway up the canal" exactly located with respect to the free surface? For better clarity, a scheme of the laser sheet might be incorporated in Fig. 11.

-Formula (3) soon after line 281 seems wrong. Why N(N-1) in the denominator, instead of (N-1)? How the reference value for the mean flow velocity was measured? The uncertainties reported in Fig. 13 could be also partially due to the fluctuations of the flow velocities, due to turbulence. Please comment in Section 5 on this possible issue.

-The discussion starting from line 318 is too short and lacks crucial details. Hence, in my opinion, it turns out to be of little use for the readers. The sentence "In order to verify the correctness of the performed numerical calculations" leaves me a bit confused. Was the numerical investigation carried out with the purpose of supporting the experimental investigation and the reliability of the measuring method or the opposite? The data reported in Fig. 14 are quite scattered, especially for large diameters: it poses crucial questions on either the reliability of the vortex device or on the reliability of the flow velocity measurements. I think that these problems should be addressed in the revised manuscript.

-Figure 14: the figure should be substantially improved. The unit of the x-axis is [mm/s], not [mm]! The continuous red line is not reported in the legend (to which diameter does it correspond?). In the text, the red dotted line is reported to be obtained by numerical simulations. However, similar to that done for the experimental data reported in the same figure, could the Authors also show the raw numerical data (i.e. the numerical data points leading to the best-fitting line)?

-I understand that the vortex frequencies reported in Fig. 14 have been experimentally measured by means of a high-dynamics pressure sensor located behind the vortex shedder. Could the Authors give some details about this crucial device? Is it in-built in the vortex flowmeter?

-In the Conclusions, the Authors state that "Only one selected fluid flow velocity was tested.". Yet, in Section 3 it is reported that 3 velocities (0.5m/s, 1.0m/s and 2.0m/s) have been numerically investigated. Please better explain or revise the conclusions accordingly.

Minor issues:

-typo in the title: please change "analyzes" to "analysis".

-the abstract should be improved. In particular, a short introductory sentence on the vortex flowmeter devices would be helpful for the non-specialist reader. As well, terms like "vortex generator" and "vortex shedder" should be shortly defined for better clarity.

-the keyword list poorly describes the main contribution. Some keywords about the trace-tracking visualization method would be useful.

-line 24: I would suggest changing "the flow of liquids" to "the velocity of fluids". As well, along with the manuscript, for better clarity, I would suggest changing the expression "fluid flow" to "flow velocity".

-line 34: the expression "flow velocity of the flow" is a repetition, please change to "flow velocity".

-line 61: typo "tested numerical" instead of "tested numerically".

-for conciseness, I would suggest merging Figures 1 and 2.

-line 104: please change "good recreation" to "good reconstruction".

-line 121: typo "elenets" instead of "elements".

-line 122: typo "cumulation" instead of "calculation".

-line 129: the sentence "the k-ε model insufficiently reflects the process of vortex formation directly downstream of the generator." is unclear. I understand that the measured vortex is the one forming upstream, not downstream of the generator. Please change or better explain.

-I understand that Figs. 5-6 are related to the numerical results by the k-ω model. Explicitly mention it in the figures captions.

-line 168: typo "Comptational" instead of "Computational".

-line 184: the expression "municipal model" sounds awkward. Perhaps the Authors wanted to express something like "universal model"? Perhaps the AutoCorrect should be disabled.

-line 188: the sentence "The experimental study was based on the proprietary method of visualizing fluid movement." is not clear. What do the Authors mean by "proprietary method"?

-lines 198-199: please revise the sentence "I want to search...", which appears naive in a journal paper.

-line 206: typo "Dancers" instead of "Tracers"?

-in Section 5 and elsewhere: please change "exposition time" to "exposure time".

-lines 312-313: the sentence is unclear, please revise.

-at line 318 I would suggest the Authors to start with a new subsection here.

-line 328: typo "vika generator"?

-Figures 14-15 should be improved. In particular, the font and markers sizes should be increased/differentiated (possibly with different colors and different shapes) for better readability.

-the use of English could be improved throughout the entire manuscript.